# Intestinal Parasites of Neotropical Wild Jaguars, Pumas, Ocelots, and Jaguarundis in Colombia: Old Friends Brought Back from Oblivion and New Insights

**DOI:** 10.3390/pathogens10070822

**Published:** 2021-06-30

**Authors:** Manuel Uribe, Esteban Payán, Jan Brabec, Juan Vélez, Anja Taubert, Jenny J. Chaparro-Gutiérrez, Carlos Hermosilla

**Affiliations:** 1Biomedical Research Center Seltersberg (BFS), Institute of Parasitology, Justus Liebig University Giessen, 35392 Giessen, Germany; mmanuel.uribe@udea.edu.co (M.U.); juan.velez@vetmed.uni-giessen.de (J.V.); anja.taubert@vetmed.uni-giessen.de (A.T.); 2CIBAV Research Group, Veterinary Medicine School, Universidad de Antioquia, Medellín 050034, Colombia; jenny.chaparro@udea.edu.co; 3Panthera, New York, NY 10018, USA; epayan@panthera.org; 4Biology Centre of the Czech Academy of Sciences, Institute of Parasitology, 370 05 České Budějovice, Czech Republic; brabcak@paru.cas.cz

**Keywords:** jaguar, puma, ocelot, jaguarundi, *Spirometra* sp., *Toxocara cati*, *Oncicola* sp., *Cystoisospora* sp., *Taenia omissa*

## Abstract

Neotropical wild felids (NWF) are obligate carnivore species present in Central and South America, and some are considered endangered due to constantly decreasing populations. NWF can become infected by a wide range of protozoan and metazoan parasites, some of them affecting their health conditions and others having anthropozoonotic relevance. Parasitological studies on NWF are still very scarce, and most data originated from dead or captive animals. On this account, the current study aimed to characterize gastrointestinal parasites of free-ranging jaguars (*Panthera onca*), pumas (*Puma concolor*), ocelots (*Leopardus pardalis*), and jaguarundis (*Herpailurus yagouaroundi*), i.e., four out of six NWF species endemic to Colombia. Fecal samples from jaguars (*n* = 10) and ocelots (*n* = 4) were collected between 2012 and 2017 as part of the Jaguar Corridor Initiative from six geographic locations in Colombia. In addition, cestode specimens were obtained during puma and jaguarundi necropsies. Scat samples were processed by standardized sodium acetate-acetic acid-formalin (SAF), sedimentation, and flotation techniques and by carbol fuchsin-stained fecal smears. Morphological evaluation of feces showed the presence of one cestode (*Spirometra* sp.), a nematode (*Toxocara cati*)*,* an acanthocephalan (*Oncicola* sp.), and one cyst-forming coccidian (*Cystoisospora-*like oocysts). Feces oocysts were submitted to a *Toxoplasma gondii*-specific PCR for species identification, but no product was amplified. The cestodes isolated from a puma and jaguarundi were molecularly characterized by sequencing cytochrome c oxidase subunit I, identifying them as *Taenia omissa* and as a *T*. *omissa* sister lineage, respectively. These results collectively demonstrate the potential role of NWF as natural reservoir hosts for neglected zoonotic parasites (e.g., *Spirometra* sp., *T. cati*) and highlight their possible role in parasite transmission to human communities. Due to public health concerns, the occurrence of these parasites should be monitored in the future for appropriate zoonotic management practices in conservation strategies and wild felid health management programs.

## 1. Introduction

The family Felidae (order: Carnivora) is currently composed of 45 recognized non-hybrid extant wild species with a worldwide distribution throughout all biomes except the Antarctic polar ice caps and insular Oceania [1,2]. All members are obligate carnivores acting as apex predators or mesocarnivores in many terrestrial ecosystems. Large wild felids serve as effective umbrella and keystone species, contributing to maintaining and regulating associated biodiversity and ecosystems where they occur [3]. Neotropical wild felids (NWF) are well-known hosts of important zoonotic protozoan parasites, such as *Toxoplasma gondii* [4,5], *Cryptosporidium* sp., and *Giardia* sp. [6,7] and are often reservoirs of hemoparasites such as *Trypanosoma cruzi* [8] and tick-borne piroplasmids such as *Babesia* sp., *Cytauxzoon felis* [9], and *Anaplasma* sp. [10]. Moreover, the presence of metazoan parasites has also been reported in non-domestic NWF, showing them as feasible hosts of gastropod-borne metastrongyloid lungworms [11,12] or *Dirofilaria immitis*, the causative agent of heartworm disease [13]. Other helminths, for instance, hookworms [14], trematodes [15], and cestodes, [13,16,17,18] have also been reported in non-domestic wild felids as well as ectoparasites like ticks, mites, and fleas [19]. The sophisticated ways in which parasite life cycles have evolved to ensure transmission involve complex interactions with vertebrate and invertebrate hosts, and parasite assemblage reflects the host’s trophic position within the food web [20]. Thus, parasite populations and communities are useful indicators of environmental stress, food web structure, and biodiversity [20,21]. The neotropics are the most diverse region with the largest number of animal species in the world [22,23], felids not being the exception. Colombia is home to six species of NWF (Table 1). Several of these species co-occur or are wholly sympatric; for example, puma, jaguarundi, and ocelot are sympatric to jaguar ranges in Colombia, but not necessarily the other way round [24,25,26].

The potential multiplicity of NWF parasite species has never been evaluated in the unique Central and South American hinge-joining key territory of Colombia. Despite numerous data on the ecology and biology of non-domestic felids in Colombia [25,27,28], little is known about free-ranging NWF-associated infectious diseases (e.g., virus, bacteria, fungi) and their parasite fauna. Additionally, parasite surveillance in natural ecological systems is an important tool to understand wildlife health, parasite biodiversity, ecology, and conservation [29]. Hence, the current study aims to present the first description of gastrointestinal parasite fauna from free-ranging jaguars, pumas, ocelots, and jaguarundis at eight sampling locations in Colombia through copromicroscopic and necropsy-based approaches on detailed morphology and further molecular identification.

## 2. Results

### 2.1. Copromicroscopical Evaluation

Parasitological evaluation of jaguar (*P. onca*) and ocelot (*L. pardalis*) faeces through basic coprological standard techniques simultaneously evidenced three metazoan parasite taxa belonging to Platyhelminthes and Acanthocephala, plus a protozoan of the phylum Alveolata (Subphylum: Apicomplexa). A high infection rate (~36%; 5/14) of cestode eggs belonging to *Spirometra* sp. was found (Figure 1a). The oval-shaped diphyllobothriid eggs corresponded to *Spirometra* sp. These asymmetric yellowish eggs showed a slightly distinct operculum at the cone-shaped pole (Figure 1b). Furthermore, we also identified golden, slightly pear-shaped ascarid-type eggs with characteristic thick-pitted eggshells (Figure 1c). Therefore, the traits of ascarid-type eggs depicted above corresponded well to the zoonotic nematode *Toxocara cati*. Additionally, pale and slightly oval eggs of *Oncicola* sp., with a delicate external membrane, were detected. Finally, un-sporulated *Cystoisopora*-like oocysts (Sarcocystidae) were also identified in jaguar and ocelot scat samples (Figure 1d,e, respectively).

### 2.2. Cestode Identification and Characterization of Rostellar Hooks

The macroscopical analyses of helminth specimens collected from the small intestine of a puma (*P. concolor*) and a jaguarundi (*H. yagouaroundi*) evidenced the presence of taeniid cestodes in both felids during necropsy procedures. Both cestodes presented a ribbon-like strobila with many proglottids. Immature and mature proglottids were wider than longer, increasing in length towards the posterior part. Additionally, two rows of hooks in a well-developed rostellum were noticed. All rostellar hooks of anterior row were larger and alternated with those of second row, which were consistently smaller (please refer to Appendix A). Armed rostellum evidences a total of 48 hooks. The basic morphological measures of large and small hooks (*n* = 24, *n* = 21, respectively) were: 282.64 µm and 205.31 µm total length (TL), 135.26 µm and 99.57 µm total width (TW), 202.89 µm and 164.62 µm blade length (BL), 135.59 µm and 97.69 µm apical length (AL), 59.28 µm and 45.34 µm guard length (GL), 47.49 µm and 38.86 µm guard width (GW), 39.32 µm and 27.46 µm blade curvature (BC), and 45.03 µm and 22.07 µm handle (HW). Along the strobila, each proglottid showed marginal alternating irregular genital pores, demonstrating the presence of a single set of reproductive organs (Figure 2). These morphological traits correspond well to the cyclophyllidean genus *Taenia*.

In order to identify the species of adult cestodes found in puma and jaguarundi, 858 bp-long fragments of cytochrome c oxidase subunit I (COI) gene of both specimens were characterized and subjected to phylogenetic analysis. Representative COI sequences of relevant *Taenia* species including all specimens reported from felids were included. The specimen from puma clustered within the lineage composed of representatives of *Taenia omissa*, while the jaguarundi isolate formed a sister lineage to them (Figure 3).

### 2.3. Toxoplasma gondii PCR

None of the jaguar nor ocelot faecal oocyst samples analyzed showed effective amplification of the *T. gondii*-specific 529 bp DNA repetitive fragment. Nevertheless, both the internal and positive controls of each test amplified normally, and negative controls never showed amplification.

## 3. Discussion

Since the vast majority of available data on wild felid parasite fauna come from captive, deceased, or highly anthropized individuals, and data on free-ranging NWF are scarce [30], the findings presented here constitute an important contribution to baseline understanding of the parasite fauna harbored by free-ranging wild felids (~67% of species) of Colombia. Agricultural expansion negatively impacts the occupancy of wild felid communities across human-modified landscapes [31,32], and these adverse anthropogenic factors may in turn influence their respective parasite communities. Thus, more frequent domestic animal–human–wildlife interface favors a plethora of infectious pathogens to emerge, spread, cross species barriers, and eventually evolve [14,33]. Here, we describe free-ranging wild felid parasites, including zoonotic parasites, heightening the importance of NWF living in human-modified landscapes and highlighting the need for appropriate zoonotic management practices in wild felid health management programs, due to public health concern and conservation.

Sparganosis is a globally distributed neglected water- and food-borne disease caused by larval stages of *Spirometra* sp. located in various human body tissues [34]. The occurrence of *Spirometra* infections across South America has been reported since the beginning of the 20th century [35]. This cestode has been previously recorded in Geoffroy’s cat (*Leopardus geoffroyi*), puma, and jaguarundi in western Paraguay [36]; the guiña (*Leopardus guigna*) in Chile [37]; and jaguar, puma, and margay (*Leopardus wiedii*) in Brazil [16,38]. Additionally, ocelots from Peru [39] and Brazil [38] have been shown to represent feasible definitive hosts of *Spirometra* sp. To the best of our knowledge, we report *Spirometra* sp. here for the first time in Colombian free-ranging jaguars and ocelots. Human cases of sparganosis have been previously reported in South America [40,41,42], but to date there is only a single six-decades-old case report from Colombia [43]. In addition, ascarid nematodes were shown to be broadly prevalent in some wild felids, and *T. cati* is the dominant parasite in some of them due to its complex life cycle, including lactogenic transmission and a wide array of paratenic hosts (e.g., rodents) [44]. We report *T. cati* in jaguars and ocelots, highlighting again the potential role of NWF in parasite transmission to local human communities. Despite the worldwide distribution of anthropozoonotic *T. cati* and its endemicity in most American countries [45], feline as well as human toxocarosis is still poorly understood in Colombian rural areas, since most studies have been conducted in urban areas, including large cities of the country [46].

Despite *T. gondii* negative molecular assays, here we describe un-sporulated *Cystoisospora*-like oocysts in wild jaguars and ocelots. Since there are at least eight *Eimeria* species described previously in felids as spurious findings, meaning that identified *Eimeria* oocysts belonged to prey animals and passed through the felid’s gut into faeces [47], it would be recommended to posteriorly identify if *Cystoisospora*-type oocysts reported here belonged to *Cystoisospora rivolta*, *Hammondia hammondi,* or *Besnoitia* spp., frequently reported as cyst-forming coccidians in domestic and wild felids [48]. Furthermore, the cyclophyllidean cestode *T. omissa*, which was firstly reported in 1910 [49], was also evidenced during necropsy procedures of the deceased puma. To date, *T. omissa* molecular data information is restricted to reports in natural intermediate hosts such as domesticated alpacas (*Vicugna pacos*) [50] and free-living red brockets (*Mazama americana*). Meanwhile, puma [51] and Eurasian lynx (*Lynx lynx*) [18] have also been reported as *T. omissa*-definitive hosts. Therefore, the present study enlarges the sequence data for this tapeworm of felids, expands the geographical distribution range of *T. omissa* to Colombia, and adds jaguarundi as a new definitive host for an uncharacterised sister linage of *T. omissa*. In comparison to faeces, carcass evaluation increases the chance of parasite detection. The copromicroscopic detection of *Taenia* sp. eggs tends to be of lower sensitivity when compared to carcass evaluation, since taeniid eggs are usually passed within mature proglottids, and show intermittent shedding analogously to other cestodes. [52]. The acanthocephalan genus *Oncicola*, consisting of twenty-four recognized species, has been circulating in South American felines for almost 9000 years [53,54]. Some scattered reports of *O. canis, O. oncicola*, and *O. venezualensis* have been reported in jaguars, ocelots, pumas, and margays across the American continent [13,55,56,57,58,59,60]. We report this parasite genus in free-ranging jaguars and ocelots in Colombia for the first time since 1968 [61].

Based on the fact that parasites are associated with retarded growth, reproductive disorders, tissue damage, inflammation, and mortality in wildlife [14], constant parasitological investigations of Colombian NWF are needed. This should be considered not only for conservation strategies and wild felid health management programs but also for public health concerns. For a better comprehension of parasite fauna infecting free-ranging NWF in Colombia, we also encourage further studies of the highly arboreal margay and the rare Northern tiger cat (*L. tigrinus*) to complement baseline data for the complete set of six endemic species reported to date in Colombian territories. Likewise, for comparative purposes, further parasitological surveys of the species included in the present study (i.e., jaguar, puma, ocelot, and jaguarundi) should be performed at a larger sample size in different biomes and seasons throughout the American continent. Finally, since the indirect life cycles of *Spirometra* sp. and *T. cati* require two to three hosts, including humans as aberrant hosts, it is desirable to analyze potential intermediate hosts (e.g., tetrapods, invertebrates, and copepods) in agricultural and semi-aquatic landscapes for a better understanding of these neglected parasites in the tropics and to delineate appropriate zoonotic health management practices to avoid human infections. Intriguing felid metastrongyloid cardiopulmonary nematodes have become spotlighted in the parasitology of wild felids [12,62]. Thus, we encourage future studies on epizootiological drivers of feline , aelurostrongylosis, angiostrongylosis, crenosomosis, gurltiosis, and troglostrongylosis in NWF [12,37,63], as these parasitoses have been discredited in populations of wild felids [11,37].

## 4. Materials and Methods

### 4.1. Study Area

The current study was conducted across the highly heterogeneous Colombian biomes of the Andean, Amazonian, and Orinoquía regions. Based on the Köppen–Geiger climate classification [64], the eight sampling geographic locations included in the present study belong to tropical monsoon (Am), tropical rainforest (Af), and tropical wet and dry climate (Aw) (see Table 2). The jaguar and ocelot faecal samples were collected from three locations in Santander, and one from Antioquia, Casanare, and Córdoba departments, respectively. Furthermore, cestode specimens (i.e., scolex, strobila, and proglottids) were collected from a deceased wild puma in Caquetá and a road-killed jaguarundi in Cundinamarca (refer to Figure 4).

### 4.2. Sample Collection and Laboratory Procedures

The current study includes free-ranging wild jaguar (*n* = 10) and ocelot (*n* = 4) faecal samples collected between 2012 and 2017 as part of the Jaguar Corridor Initiative, a conservation and monitoring program carried out by the Panthera organization across the jaguar’s range [65]. Collected faeces came from direct sampling sites of trails that were regularly monitored by trap cameras (refer to Figure 5). Faeces were identified by associated tracks and followed the general features and morphometric characteristics of wild felid depositions [66]. The traditional ecological knowledge of locals was also very helpful for monitoring and sampling these reclusive individuals [67]. Once identified in the field, well-formed faeces were dry preserved and fixed in 70% EtOH until subsequent copromicroscopic evaluation, as recommended for challenging tropical environments [68]. Furthermore, we collected a cestode sample serendipitously during the necropsy of a young road-killed wild jaguarundi male in Cundinamarca. The cestode specimen was carefully extracted from the ileum. Additionally, free cestode proglottids and whole tapeworms firmly attached by their scolex to the jejunum mucosa of an adult deceased female puma at Caquetá were collected. Cestode specimens were gently rinsed in physiological buffered saline solution (PBS) and thereafter preserved in 96% EtOH until further molecular evaluation. All sampling procedures were performed in agreement with the Guidelines of the American Society of Mammologists for the Use of Wild Mammals in Research and Education [69,70], the EU Directive 2010/63/EU, and the final approval of the Ethics Committee for Animal Experimentation of the Universidad de Antioquia (AS No. 132) under collection permit No. 0524 of 2014 (IDB0321), Colombia.

#### 4.2.1. Basic Copromicroscopic Analyses

Since there is no single copromicroscopic method to diagnose all parasitic stages concomitantly, the jaguar and ocelot faeces examination was performed by means of the following qualitative techniques for cysts, oocysts, eggs, and parasite larvae detection to optimize data collection: modified sodium-acetate aceticacid formaldehyde (SAF) technique [71], simple sedimentation technique [72], zinc sulfate centrifugal flotation technique, and fast carbol-fuchsin stained faecal smears [73]. The parasitic specimens were identified through morphometry under an Olympus BX53 (Olympus Corporation, Tokyo, Japan) semi-motorized direct light microscope (100×, 400×, and 1000×) equipped with an Olympus DP74 (Olympus Corporation, Tokyo, Japan) digital camera using the *cellSens* standard imaging software (Olympus Corporation, Tokyo, Japan). The parasites’ identification was based on general morphology, shape, size, and color, according to Deplazes et al. (2016) [74].

#### 4.2.2. Molecular Phylogenetics

Adult cestodes obtained from the jejunum of the female puma and the ileum of the male jaguarundi were carefully removed from the epithelium and/or lumen of the small intestine, trying to preserve the scolex, strobila, and proglottids integrity. The obtained helminths were first photographed using a stereomicroscope (Nikon SMZ25R, Tokyo, Japan). Amplification of partial cytochrome c oxidase subunit I (COI) was achieved with primers JB3 [75] and Cox1R [76] using Phusion High-Fidelity DNA Polymerase (New England Biolabs, Inc., Ipswich, USA) and the following cycling conditions: 35 cycles of 10 s at 98 °C, 15 s at 60 °C, and 50 s at 72 °C. PCR products were gel-checked, purified with Exonuclease I and FastAP alkaline phosphatase (Thermo Fisher Scientific, Waltham, USA), and directly Sanger-sequenced at SeqMe (Dobříš, Czech Republic). Contiguous gene sequences were assembled and inspected for errors in Geneious 7.1.9 (http://www.geneious.com, accessed on 3 June 2021 [77]). COI coding sequences were aligned using MAFFT’s L-INS-i [78] translational align of Geneious. The phylogenetic tree was estimated by maximum likelihood in IQ-TREE 1.6.5 [79]. The best-fitting model of nucleotide evolution was chosen according to the corrected Akaike information criterion in IQ-TREE ([80,81]) and nodal supports estimated through running 100 standard nonparametric bootstrap replicates.

#### 4.2.3. Faecal DNA Isolation and *Toxoplasma gondii* Molecular Evaluation

Total DNA isolation was performed using the Class II type B2 BSC and the DNeasy Blood & Tissue Kit (Qiagen, Venlo, Netherlands) following manufacturer’s instructions. A 200- to 300-fold repetitive 529 bp DNA fragment conserved among 60 strains and more sensitive than B1 gene was used for *T. gondii* molecular detection. Amplification of repeated fragments was performed using Toxo4 and Toxo5 primers set under previously described conditions [82]. Tachyzoites of the *T. gondii* RH- and ME49 strain were used as positive DNA controls.

## Figures and Tables

**Figure 1 pathogens-10-00822-f001:**
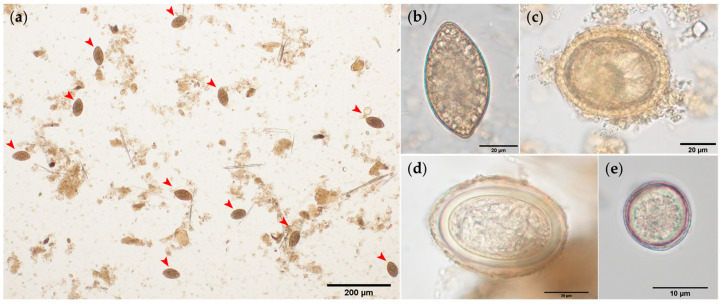
Illustrations of parasite eggs detected in faecal samples of free-ranging jaguars and ocelots: (**a**) High number of *Spirometra* sp. eggs; (**b**) Single *Spirometra* sp. egg *(*60.72 µm × 33.38 µm); (**c**) Non-embryonated *Toxocara cati* egg (63.86 µm × 53.43 µm) carrying a zygote; (**d**) *Oncicola* sp. egg (64.30 µm × 46.68 µm); (**e**) Un-sporulated *Cystoisospora-*like oocyst (12 µm × 12 µm; Sarcocystidae). Scale-bars: (**a**) 200 µm; (**b**–**d**) 20 µm; (**e**) 10 µm.

**Figure 2 pathogens-10-00822-f002:**
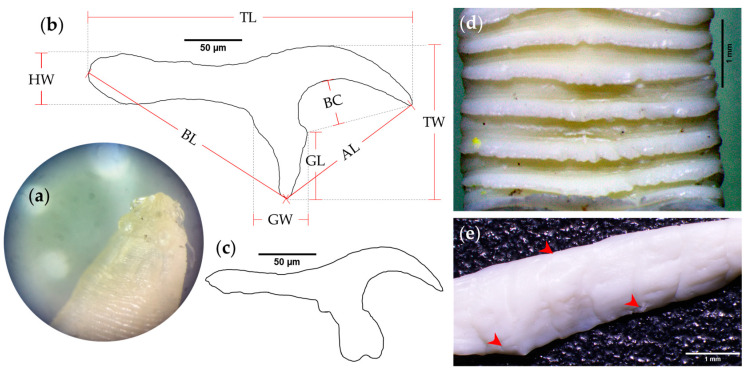
Morphological traits of putative *Taenia omissa* specimens. (**a**) Scolex photograph of *T. omissa* obtained from puma; (**b**) Large and (**c**) small rostellar hooks outline drawings of adult *Taenia* sp. specimen from jaguarundi gastrointestinal tract (ileum); (**d**) jaguarundi, and (**e**) puma cestodes ribbon-like strobila (red arrows indicate genital pores). TL: total length, TW: total width, BL: blade length, AL: apical length, GL: guard length, GW: guard width, BC: blade curvature, HW: handle width. Scale-bars: (**b**,**c**) 50 µm; (**d**,**e**) 1 mm.

**Figure 3 pathogens-10-00822-f003:**
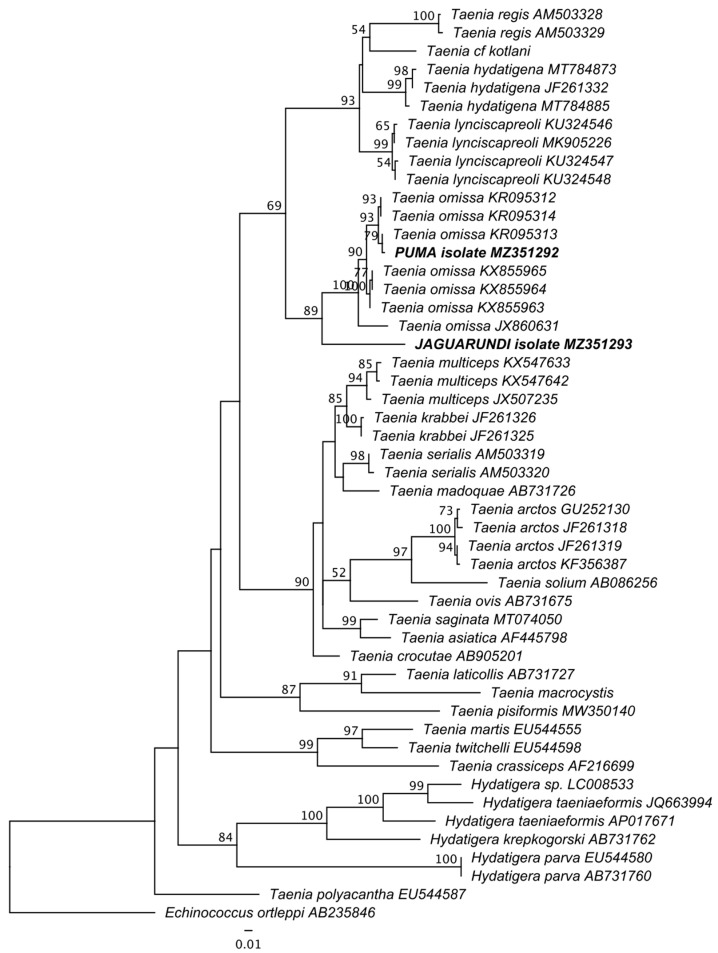
Phylogenetic position of *Taenia* sp. isolates obtained from puma and jaguarundi. Maximum likelihood tree from IQ-Tree based on cytochrome c oxidase subunit I gene sequences analyzed as single partition using GTR + F + I + G4 model selected according to corrected Akaike information criterion. Nodal values show standard bootstrap supports above 50 (100 replicates). Specimens collected from puma and jaguarundi are shown in bold. GenBank accessions are given after taxa names. The branch length scale bar indicates number of substitutions per site.

**Figure 4 pathogens-10-00822-f004:**
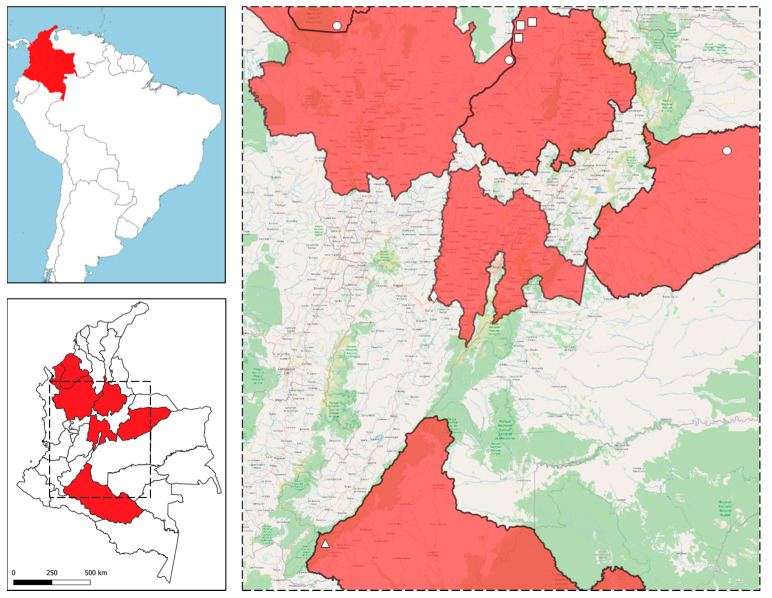
Geographical locality of sampled free-ranging neotropical wild felids (NWF).

**Figure 5 pathogens-10-00822-f005:**
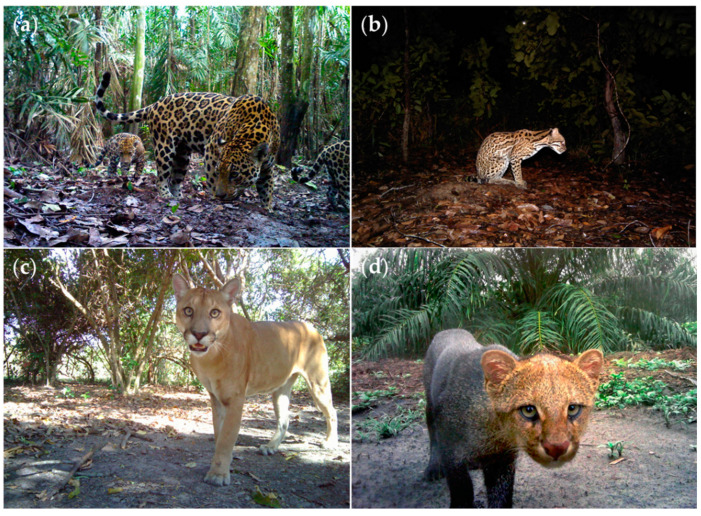
Camera trap-images of monitored free-ranging neotropical wild felids (NWF) of the present study: (**a**) jaguar female (*Panthera onca*) with two cubs; (**b**) adult ocelot (*Leopardus pardalis*); (**c**) puma (*Puma concolor*); (**d**) jaguarundi (*Herpailurus yagouaroundi*).

**Table 1 pathogens-10-00822-t001:** Neotropical wild felid (NWF) species of Colombia.

			Risk Classification		
Genus	Species	Common Name	CITES ^a^	UICN ^b^	National ^b^
*Herpailurus*	*yagouaroundi* *	Jaguarundi, Eyra cat	Appx II	LC	NE
*Leopardus*	*pardalis* *	Ocelot	Appx I	LC	NE
*Leopardus*	*wiedii*	Margay, Tree ocelot	Appx I	NT	NE
*Leopardus*	*tigrinus*	Northern tiger cat	Appx I	VU	NE
*Panthera*	*onca* *	Jaguar	Appx I	NT	VU
*Puma*	*concolor* *	Puma, Cougar	Appx I	LC	NE

* Species included in the current study. ^a^ All appendix I species are threatened with extinction. ^b^ LC: least concern; NT: near threatened; VU: vulnerable; NE: not evaluated.

**Table 2 pathogens-10-00822-t002:** Detailed sampling areas and climate classification.

Department	Municipality	Sampling Location	Climate ^a^	Sample Type
Antioquia	Yondó	Ciénaga de Barbacoas	Am Af	Feces ^o^
Caquetá	San José del Fragüa	Puerto bello	Af	Metazoan ^Δ^
Casanare	Hato Corozal	La Chapa	Am	Feces ^o^
Córdoba	Puerto Libertador	La Esmeralda	Af	Feces ^o^
Cundinamarca	-	-	Aw	Metazoan ^◊^
Santander	El Hato	Las Pampas	Af	Feces ^□^
Santander	Puerto Wilches	Las Palmas	Am	Feces ^□^
Santander	Puerto Wilches	Caño Limón	Am	Feces ^□^

^a^ Köppen–Geiger Am: Tropical monsoon; Af: Tropical rainforest; Aw: Tropical wet and dry. ^o^ Jaguar (*Panthera onca*) faeces. ^□^ Ocelot (*Leopardus pardalis*) faeces. ^Δ^ Puma (*Puma concolor*) collected helminths. ^◊^ Jaguarundi (*Herpailurus yagouaroundi*) collected helminths.

## Data Availability

The putative *Taenia omissa* sequences obtained from puma (*Puma concolor*) [PUM_CPB_Fem_20] and jaguarundi (*Herpailurus yagouaroundi*) [YAG_CG_Male19] were deposited in GenBank database (National Center for Biotechnology Information, NIH, Bethesda, USA) and are available at ENA (European Nucleotide Archive) in Europe and the DDBJ (DNA Data Bank of Japan) under accession numbers MZ351292 and MZ351293, respectively.

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
