# Peer review of "Intestinal Parasites of Neotropical Wild Jaguars, Pumas, Ocelots, and Jaguarundis in Colombia: Old Friends Brought Back from Oblivion and New Insights"

_pathogens, 2021, doi:10.3390/pathogens10070822_

Round 1

Reviewer 1 Report

Very interesting study, especially in the context of protected and endangered species/host. The study presents the new data on the felid parasites from the area of Southern America. 

The article was enriched with an interesting supplementary video, although I do not know how the authors did it.

I am of an opinion that the article fits into scope of Pathogens

Comments

Please add authors and date to the names of all species (hosts, parasites), eg. Panthera onca  (Linnaeus, 1758).

Abstract: Please remove the information about Cryptosporidum - this entry is misleading; currently the authors have not found this parasite, besides, further in the article (Material and methods, Results) there is no mention of it.

Results: I propose to specify the entry that the found parasites (Spirometra, Toxocara, Onicola, Cystoisospora) were found simultaneously in the examined jaguars and ocelots (faecal samples).

Results, 2.2. … measures of large and small hooks (n = 24, respectively)..:  at the same time for large and small ?, please check and correct.

Discussion: The first and second chapters deal  mainly with zoonotic species, the third with other parasites, and the fourth is a summary.

I suggest changing/reformatting the third chapter on acanthocephalan, apicomplexan and Taenia omissa. Rather, it would be better to start with Apiciomplexa, then Cestoda and finally Acanthocephala.

Figure 4, line 225: location – should be locality; location – habitat, site; locality – geographic locale of the external environment.

References

  1. Please check the Parasites and Vectors (4, 9, 11) abbreviation.
  2. Please correct the capitalization (7, 8, 27, 30, 33, 40, 63, 64, 68, 73, 77)
  3. Article (chapter) “Reperant, L.A.; Cornaglia, G.; Osterhaus, A.D.M.E. The Importance of Understanding the Human–Animal Interface” is from “One Health: The Human–Animal–Environment Interfaces in Emerging Infectious Diseases” – please correct.
  4. Moulinier, R.; Martinez, E.; Torres, J.; Noya, O.; de NOya, B.A.; Reyes, O. Human proliferative sparganosis in 434 Venezuela: report of a case. J. Trop. Med. Hyg. 1982, 31, 358–63 – should be … 358-363.
  5. “López-Osorio, S.; Penagos-Tabares, F.; Chaparro-Gutiérrez, J.J. Prevalence of Toxocara spp. in dogs and cats in 447 South America (excluding Brazil). Advances in Parasitology…. . - Advances in Parasitology is a journal, please correct.
  6. Amin, O.M. Classification of the Acanthocephala. Folia Parasitol. (Praha) - please delete “(Praha)”.
  7. BOTERO, D.; GOMEZ, J.J. The first case of sparganosis … - please correct the capitalization.
  8. Patton, S.; Rabinowitz, A.; Randolph, S.; Strawbridge Johnson, S. A coprological survey … - should be …Johnson, S.S. ….
  9. Palmer, J.P.S.; Dib, L.V.; Lobão, L.F.; Pinheiro, J.L.; Ramos, R.C.F.; Uchoa, C.M.A.; Bastos, O.M.P.; Silva, M.E.M.; 467 Nascimento, J.L. Do; Pissinatti, A.; et al. Oncicola venezuelensis (Marteau, 1977) (acanthocephala … - should be Acanthocephala … .

Author Response

Dear Reviewer;

First, we would like to thank you for the manuscript comments and suggestions, all of them in favor to improve it. Regarding the 3D modeling of the rostellar hooks, cellSens™ standard imaging (Olympus) and Rhinoceros 3D software were used for morphometry, 2D shape lifting and final 3D modeling of the silhouette. Similarly, AutoCAD, Houdini, Paint 3D, Autodesk Maya, or any generic 3D modelling software could also work.

We do not recommend using the taxonomic authors of the scientific names throughout the text because it will confuse and dilute attention to the results in question. Specially in a multi-species (Vertebrates, protozoan, and helminths) paper. E. g., yagouaroundi Desmarest (1816), is a junior synonym and placed in Herpailurus sp. by Weigel (1961). The genera Puma sp., which was recently used for the species, then revoked lately was set by E. Geoffroy Saint-Hilaire. None of this is the focus of the current paper.

The mention of Cryptosporidum sp. in the abstract were just indicate the diagnostic target of the carbol fuchsin-stained in faecal smears, but since it could be confusing it was removed as you recommended. The amendment of simultaneously copromicroscopical findings in both analyzed felids was carried out as equal as the rearrangement of third paragraph in discussion. Precise number of measured large and small rostellar hooks were verified. General discussion structure was modified to describe Apicomplexa, then Cestoda and finally the Acanthocephala findings. The change of “location” by “locality” at Figure 4 was performed too. Additionally, extensive review and correct quotation of full references list was made.

PS: Attaches below you will find a short cover letter which details the changes.

Best regards.

Reviewer 2 Report

Dear Authors,

It is this reviewer opinion that the ms need few revisions. Points to be addressed are listed below.

Line 54. “Moreover, the presence of metazoan parasites has also been reported in non-domestic NWF, showing them as feasible hosts of gastropod-borne metastrongyloid lungworms [11].” Ref. n. 11, i.e. Dimzas et al., 2020, describes the presence of feline lungworms in their intermediate hosts in an area of wildcat and domestic cat sympatry. References should be implemented with at least one article describing the presence of these nematodes in wild and domestic felids (e.g. https://pubmed.ncbi.nlm.nih.gov/32057385/).

Moreover, though out of the scope of this ms, the authors should underline and discuss the role of respiratory nematodes which have come in the spotlight in the parasitology of wild felids.

See, discuss and cite in the discussion section these references which contains many information which could be used for this scope:

https://pubmed.ncbi.nlm.nih.gov/32351980/

https://pubmed.ncbi.nlm.nih.gov/32057385/

Although I’m not a mother-tongue, my feeling is that the English should be slightly revised. For instance at lines 194-197 the correlation between the intermittent egg shedding and the low sensitivity of the copromicroscopic techniques should be better specified; at lines 82, 195, 258 and 259 Flotation, SAF, sedimentation and faecal smears should be defined “copromicroscopic” techniques and not coproscopical, (here and throughout the text); line 264: “parasitic elements” would be more appropriate than “parasitological findings”

Line 198: “certain gastrointestinal parasites”, please specify which one.

Author Response

Dear Reviewer;

First, thank you for such clear suggestions and text comments to improve the manuscript. Regarding the introduction background we decide to change ref Dimzas et al. 2020 (10.1186/s13071-020-04213-z), since as you indicate, it describes metastrongyloid nematodes of gastropod hosts in a Greek area of sympatric wild/domestic felids. Thus, since current manuscript focused on wild felids we include two papers which describe lungworms in Chilean Leopardus guigna, and tropical wild felids (i. e., 10.1016/j.ijppaw.2020.07.013, and 10.1007/s00436-016-5134-y).

The text was subjected to an extensive review, syntax double-check, illative use of English, and accurate reference quoting. Misleading phrases were rewording and edited to clearly develop the main idea. The word “coproscopical” were change throughout the manuscript to “copromicroscopic” as equal as “parasitological findings” by “parasitic elements”.

PS: Attached below you will find a short cover letter which details the changes.

Warm regards.

Reviewer 3 Report

This study describes the findings of various tests for intestinal parasites in an opportunistic sample of faeces or intestinal contents of neotropical wild felids in Columbia. Despite a very small sample size (given the difficulty in obtaining samples from these animals in the wild) the authors did find 5 parasites - Toxocara cati, Taenia omissa, Spirometra spp., Cystoisospera and Onsicola spp.. Their findings are worthy of publication since there no prior data from NWF in Columbia, given the near threatened and vulnerable classification of these species, and since some of identified parasites are zoonotic. Overall the study is well-written and the findings clear.

I don’t believe that the title is appropriate for the study. I would recommend removing “Old friends brought back from oblivion and new insights” as I think this is misleading.

The limitations of the study should be included in the discussion. For example, the number of animals sampled relative to the estimated population of the NWF species in Columbia should be addressed. While the reviewer appreciates how very challenging it is to obtain faecal samples from these species in the wild, the sample size is nonetheless small, and as such whether or not it represents the wider population is unclear. Additionally, the authors should mention ova or oocysts of various parasites are only shed intermittently in the faeces of infected hosts and thus more pathogens may be present that were not detected (particularly given that presumably a single faecal sample was only obtained from each individual cat). It should also be noted as a limitation that faeces was not collected at the time of the puma and jaguarundi necropsies; rather only visible parasites were sampled from the small intestine.

The authors also only used some diagnostic methods, which would have skewed their findings. For example, although Giardia is mentioned in your introduction, it does not appear that PCR for this pathogen was performed. Tritrichomonas fetus, which is becoming an increasing concern in domestic cats, was also not mentioned.

Other specific comments are listed below:

  • Line 17 – I think “some are considered” is better terminology here than “some still considered”
  • Line 21 – should be “aimed” rather than “aims”
  • Line 178 – please reword this sentence – I presume you mean that you “report this parasite genus ….for the first time since 1968.”
  • Line 271 – please reword to “small intestinal epithelium and/or the lumen…”. Please remove the words “of the organ”
  • Line 303 – you state here that this research involved no funding, but then mention numerous funding sources in the acknowledgements section. Please clarify
  • Please 326 – please reword to clarify that “No funder had any role in the design of the study …”
  • Line 476 – reference 57 appears incomplete

Author Response

Dear Reviewer;

First, we would like to thank you for such a precise and clear text corrections to improve the manuscript, and for considering the huge sampling effort of those elusive animals. The wording of misleading phrases was re-checked and edited to clearly develop the main idea as equal as an extensive review of accurate reference quoting.

Precisely due to the small simple size, there is little use for a comparison with total population numbers per species. For comparative purposes, further parasitological surveys of the species included in the study (i.e., jaguar, puma, ocelot, and jaguarundi) should be performed in larger sample size throughout the American continent. Furthermore, since there are not published reports of total numbers of wild cat species for Colombia, except for one sound estimate for jaguars: 16,598 (11,724±21,311) as per JÄ™drzejewski et al. 2018 (10.1371/journal.pone.0194719). We would like to keep the current manuscript title. With “old friends” we refer to Oncicola sp. (Circulating ~9000 years-ago in NWF) and T. omissa, parasites report in Colombia for the first time since 1968 and 1910, respectively. In the same way we pretend to “brought back from oblivion” zoonotic neglected parasite infections like sparganosis with a single six-decades-old human case report, and the poorly understood toxocariasis in rural areas of the Americas.  General discussion was rearranged and "summary" fourth paragraph edited in section 3. Unfortunately, through anyone of the 4 copromicroscopic techniques we were able to identify Giardia sp. cysts nor Tritrichomonas foetus trophozoites. Fresh faeces collection for optimum results is a hard achievement. To the best of our knowledge there are no T. foetus reports in wild felids to date (10.1371/journal.pone.0246957, 10.3390/pathogens9030203, and 10.2478/jvetres-2019-0072) but future coproELISA and molecular approaches for T. foetus, Tetratrichomonas felistomae and  Giardia sp. detection in wild felids will be considered.

In the abstract sentence “some still considered” was changed by “some are considered” as suggested as equal as “aims” by “aimed”.  The words “of the organ” were removed. The funding and acknowledgement issues were amended. We clarify that no funder had any role in the design of the study at Conflicts of Interest section.

PS: Attached below you will find a short cover letter which details the changes.

With kind regards.
